# ATTACKING AND SECURING MASKING SCHEME FOR TEE-BASED MODEL PROTECTION

## ABSTRACT

Deep learning (DL) models are being increasingly adopted across a wide range of applications. Many inference models are deployed on edge devices to enable efficient and low-latency computation. However, such deployment exposes security risks, including the potential leakage of model parameters. To address these security risks, several researchers have proposed protection schemes for deployed models based on Trusted Execution Environments (TEEs).

In this paper, we analyze a common weakness of existing TEE-based protection schemes, namely the insecurity of the masking mechanism. Existing masking schemes not only provide limited security guarantees but also incur high computational and storage complexity. Motivated by these inherent weaknesses, we develop a targeted differential attack that can accurately recover the parameters of linear layers in ReLU-based neural networks. Furthermore, we propose an improved masking scheme that achieves higher security and efficiency by generating substantially more mask combinations under the same computational cost, thereby considerably strengthening TEE-based model protection.

## 1 INTRODUCTION

Over the past few decades, the field of machine learning has witnessed remarkable and unprecedented advancements, progressing at a pace that has often surpassed prior expectations. In certain domains, its performance has not only matched but even exceeded human-level capabilities. For example, in areas such as image recognition, natural language processing, and speech recognition, state-of-the-art machine learning models have achieved extraordinary accuracy and robustness. In medical imaging, deep learning techniques have demonstrated superior diagnostic performance in detecting specific pathologies compared to experienced clinicians. Similarly, in complex strategic games such as Go and chess, systems like AlphaGo have decisively outperformed world champions(Silver et al., 2016). More recently, large-scale language models such as ChatGPT have demonstrated remarkable capabilities in natural language understanding and generation, further underscoring the rapid and transformative progress of machine learning. These achievements underscore the rapid evolution and powerful potential of machine learning, continually reshaping the boundaries of what is considered possible in artificial intelligence research.

However, training a well-performing neural network often entails substantial costs, including considerable time, financial resources, and human effort. The optimization of large-scale models typically requires access to extensive computational infrastructure, such as high-performance GPUs or distributed clusters, and prolonged training times to converge to satisfactory performance. Additionally, the process of designing appropriate architectures, tuning hyperparameters, and curating high-quality datasets demands significant expertise and manual intervention.

Trained neural networks are often regarded as a form of competitive advantage and valuable intellectual property(Wenskay, 1990). Consequently, they are typically kept secret to protect the substantial investment of resources, expertise, and data that went into their development. Organizations tend to treat the parameters and architecture of their models as proprietary assets, restricting access to prevent unauthorized use, replication, or reverse engineering.

Given that cloud-based services often suffer from high latency and instability, manufacturers increasingly prefer to deploy model inference services on edge devices(Zhang et al., 2024). These edge devices are typically equipped with high-performance accelerators such as Graphics Processing Units (GPUs), Neural Processing Units (NPUs)(Tan & Cao, 2022). However, these edge devices are often considered untrusted environments, as attackers may gain full control over them. In such scenarios, adversaries can access the device's memory, intercept intermediate computations, or even manipulate the hardware and software stack. This level of access poses serious threats to the confidentiality and integrity of deployed deep neural networks, making them vulnerable to model extraction, parameter tampering, and other security breaches. Therefore, ensuring the robustness and security of DNN inference on untrusted edge devices has become a critical research challenge.

Existing approaches to protecting DNNs on edge devices can be broadly categorized into two main classes: cryptography-based methods(Al-Rubaie & Chang, 2019) and Trusted Execution Environment (TEE)-based methods(Sagar & Keke, 2021). Among the two solutions, the TEE-based approach is generally considered more practical for real-world applications with stringent performance requirements, as it incurs an acceptable computational overhead— typically less than $10\times$(Tramer & Boneh, 2018) — and does not compromise model accuracy. In contrast, cryptographic solutions often introduce substantial overhead, sometimes reaching up to $1000\times$ (Ng & Chow, 2021), and may also lead to a drop in accuracy. In this paper, we primarily focus on TEE-based protection schemes.

We analyze the security of existing TEE-based protection schemes(Hou et al., 2021; Zhou et al., 2023; Shen et al., 2022; Sun et al., 2023; Zhang et al., 2024) and identify a common weakness in their masking strategies: masks must be precomputed, which incurs substantial computational and storage overhead, and once the precomputed masks are exhausted, mask reuse becomes inevitable, leading to security vulnerabilities. By leveraging the insecurity of the precomputed masking scheme, we develop a differential attack that can compromise the confidentiality guarantees of TEE-based protected models. The attack is highly effective against shallow networks. Although the attack fails to directly reconstruct a model that closely matches very deep architectures, the recovered model can be refined via a small amount of post-recovery training to achieve high similarity with the original network.

To improve the security of model obfuscation solutions, we propose an enhanced masking scheme that increases the independence among mask components. As a result, both the efficiency of model inference and the security of the protection scheme are improved.

To assess the practicality of our attack, we experiment on a custom residual network. Experimental results indicate that the proposed attack is capable of accurately reconstructing this shallow architecture; the recovered model attains more than 98% similarity to the original model in terms of inference outputs. We also evaluated the security of the new masking scheme, and the results demonstrate that it achieves a substantial improvement in security without incurring any additional computational cost.

The contribution of this paper are as follows:

- We analyze a common weakness in existing TEE-based model protection schemes and propose an attack that exploits this weakness. The attack can recover shallow networks with very high fidelity. For deep networks it substantially reduces the amount of training data required for post-recovery refinement.

- To assist this attack, we proposed a new masking scheme. This scheme is able to generate more mask variants with the same computational cost as before, thereby improving both the security and efficiency of TEE-based protection schemes.

- We experimentally validate the feasibility of our attack on a shallow network, the attack is able to recover a model that achieves up to 98% similarity with the original network.

## 2 RELATED WORK AND BACKGROUND

### 2.1 EXECUTION ENVIRONMENT

**Untrusted Execution Environment.** Both Graphics Processing Units (GPUs) and Neural Processing Units (NPUs) have become indispensable accelerators for neural network inference due to their ability to deliver extremely high computational speed. By exploiting massive parallelism and hardware-level optimizations, these processors can execute large-scale matrix operations and convolutional computations with remarkable efficiency. Their high throughput significantly reduces inference latency, enabling real-time performance in applications such as computer vision, natural language processing, and speech recognition. As a result, GPUs and NPUs provide the computational foundation that makes modern AI models practical in both cloud and edge environments. However, while their performance advantages are widely recognized, the security aspects of GPUs and NPUs are often overlooked. Lee et al. (2014); Sorensen & Khlaaf (2024); Zhu et al. (2021) et al. exploited security vulnerabilities in GPUs to recover a large amount of sensitive information.

**Trusted Execution Environment(TEE).** A Trusted Execution Environment is a secure enclave that stores and process sensitive data. Popular TEE implementations include Intel SGX(McKeen et al., 2013), AMD SEV (Kaplan et al., 2016), and TrustZone(Alves, 2004). In CNN model protection, TEE provides a physical isolation scheme in the hardware devices that separates memory into normal(untrusted) world and the secure(trusted) world. The normal world can interact with the secure world through a secure monitor call interface(Ye et al., 2018). However, TEE typically suffer from limited computational resources and constrained memory capacity. As a result, computationally intensive

operations are usually executed outside the TEE. In this paper, we treat the TEE as a black box, meaning that neither regular users nor adversaries can observe its internal state or extract side-channel information. Only the inputs and outputs of TEE are assumed to be visible.

## 2.2 TEE-BASED MODEL PROTECTION SCHEME

To simultaneously ensure the security of the model inference process while maintaining high efficiency, researchers have proposed a TEE-based model protection scheme. Since linear operations constitute more than 95% of the model inference workload, securely offloading these computations to GPUs can substantially accelerate the inference process. Regarding the secure offloading of these computations to GPUs, numerous approaches have been proposed in recent studies (e.g., (Hou et al., 2021; Shen et al., 2022; Zhou et al., 2023; Sun et al., 2023; Zhang et al., 2024)), aiming to protect model confidentiality and integrity while maintaining high inference efficiency. These approaches generally follow a similar workflow: during a pre-processing stage, the parameters of the linear layers are obfuscated, and the obfuscated parameters are then offloaded to the GPU. The GPU performs the linear layer computations, and the results are returned to the TEE, where the true parameters are reconstructed. The main differences among these approaches lie in the specific methods used to obfuscate the parameters of the linear layers. Figure 1 illustrates the TEE-based model protection scheme.

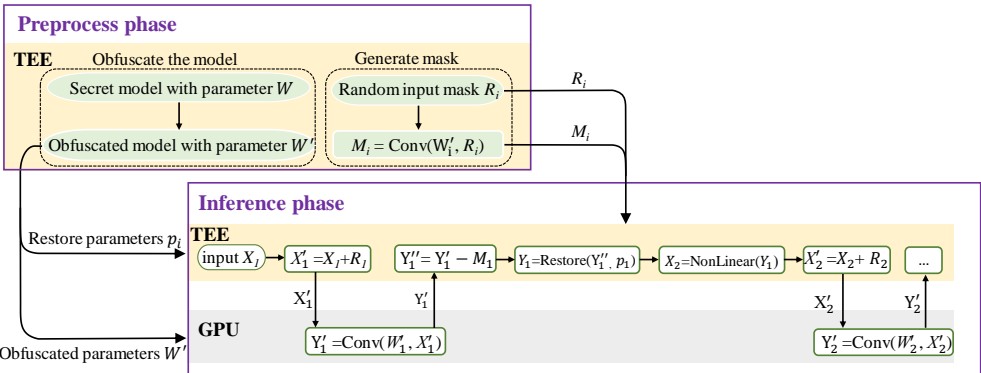

Figure 1: TEE-Based Model Protection Scheme.

It is worth noting that in these approaches, a mask is applied to the input of each linear layer to provide additional protection. Since the combination of a linear layer's inputs and outputs(inputs of the subsequent layer) can be leveraged to reconstruct its parameters through straightforward linear algebraic methods. During the preprocessing stage, such schemes generate random input masks for each linear layer and compute the corresponding output masks.

This pre-masking strategy introduces two significant limitations. First, during the precomputation stage, random input masks must be generated in advance for each linear layer, and the corresponding output masks are computed. This process is also computationally intensive and incurs substantial overhead. Second, to ensure system availability, a large set of input and output masks must be precomputed. Once depleted, masks are inevitably reused, which introduces significant security risks. Attackers can identify repeated masked input-output pairs and, by applying differential attacks to eliminate the effect of the masks, construct linear equations to recover the network parameters. This attack methodology is described in detail in Section 3.

## 2.3 THREAT MODEL

**Adversary's Capabilities.** The adversary is assumed to have full knowledge of the target model's architecture and the public pre-trained parameters. During model inference, all data outside the TEE is assumed to be visible to the adversary, including intermediate values transmitted between the TEE and external accelerators such as GPUs. Only the data residing and processed within the TEE remains confidential.

**Adersary's Goal.** The adversary ultimately aims to reconstruct a model that shares the same architecture and is functionally as well as parametrically identical to the original. It should be noted that the adversary's goal is not merely to obtain a model with high predictive accuracy, but rather to recover one with maximal similarity to the original network in both parameters and functionality.

## 2.4 Target Neural Network

In our setting, the target neural network under attack employs the ReLU activation function, which is y = ReLU(x) = max(x,0). We regard the combination of convolution, batch normalization, and ReLU activation as one logical layer of the convolutiuonal neural network(CNN). Furthermore, the activation probability of each ReLU unit in the target network is not extreme, i.e., it does not approach 0 or 1.

## 2.5 Convolution and Batch Normalization as a Single Linear Layer

Consider a convolutional layer followed by a batch normalization (BN) layer in a neural network. Let the $i$-th convolutional layer have weights $W_i$ and biases $b_i$, with input $X_i$:

$$Y_i = W_i * X_i + b_i,$$

where $*$ denotes the convolution operation. The subsequent BN layer with parameters $\gamma_i, \beta_i, \mu_i, \sigma_i$ produces:

$$Y_{ibn} = \gamma_i \frac{Y_i - \mu_i}{\sigma_i} + \beta_i.$$

We can combine the convolution and BN into an equivalent linear layer with composite parameters $W_c$ and $b_c$:

$$W_c = \frac{\gamma_i}{\sigma_i} W_i, \quad b_c = \frac{\gamma_i}{\sigma_i}(b_i - \mu_i) + \beta_i,$$

so that

$$Y_{ibn} = W_c * X_i + b_c.$$

This formulation demonstrates that a convolutional layer followed by a BN layer can be treated as a single linear layer with equivalent weights $W_c$ and biases $b_c$. This abstraction is useful for analytical purposes and for attacks that aim to recover linear parameters efficiently. In this work, we refer to the combination of a convolutional layer, batch normalization, and a ReLU activation as a single layer of the neural network.

## 3 Attack against Existing TEE-based DNN Protection Schemes

Our attack targets a fundamental weakness in TEE-based protected inference: under a burst of inference requests, certain intermediate activations are masked with the same random values. In this section, we present our proposed attack targeting TEE-based protection schemes. Specifically, we collect the input and output of each layer and recover the composite parameters of the convolution and batch normalization layers by constructing and solving algebraic equations. In Section 3.1, we analyze the data before and after masking to determine which observations can be used for parameter recovery. In Section 3.2, we take the first recovery round as an example to illustrate how masked data can be utilized to recover the parameters of a neural network. In Section 3.3, we describe how to recover parameters in subsequent layers. In Section, we describe how to detecting data that share the same mask.

### 3.1 Data Analysis

**Observation 1** (Sparsity of ReLU Outputs in CNNs). *In convolutional neural networks (CNNs), the ReLU activation function, defined as $y = ReLU(x) = \max(x, 0)$, produces a significant number of zero outputs. That is, for many input activations $x$, the ReLU function outputs $0$, resulting in sparse activation maps.*

Based on Observation 1, when two ReLU outputs within the same layer are protected with the same mask, some components of their outputs will take identical values equal to the mask itself, since the original values of these components were zero before masking. Consequently, if identical values are observed across different outputs of the same ReLU layer, it implies that the corresponding neurons share the same mask.

**Definition 1** (Collision Triplet). *In a TEE-protected inference process, a* collision triplet *refers to three intermediate activations from the same layer, each corresponding to different inputs, that are masked with the same random mask.*

Although the mask conceals the true values, certain data characteristics are preserved in inputs that share the same mask. By analyzing these characteristics in the masked data, we can infer the underlying true values. Collision triplets serve as a highly effective tool for facilitating our analysis. Using collision triplets, the values at corresponding indices can be compared to identify which components were activated prior to masking, enabling the computation of the true differentials.

When a collision triplet is collected, we analyze its data characteristics in order to eliminate the effect of the mask. Specifically, the analysis leads to the following four possible cases, as illustrated in Figure 2.

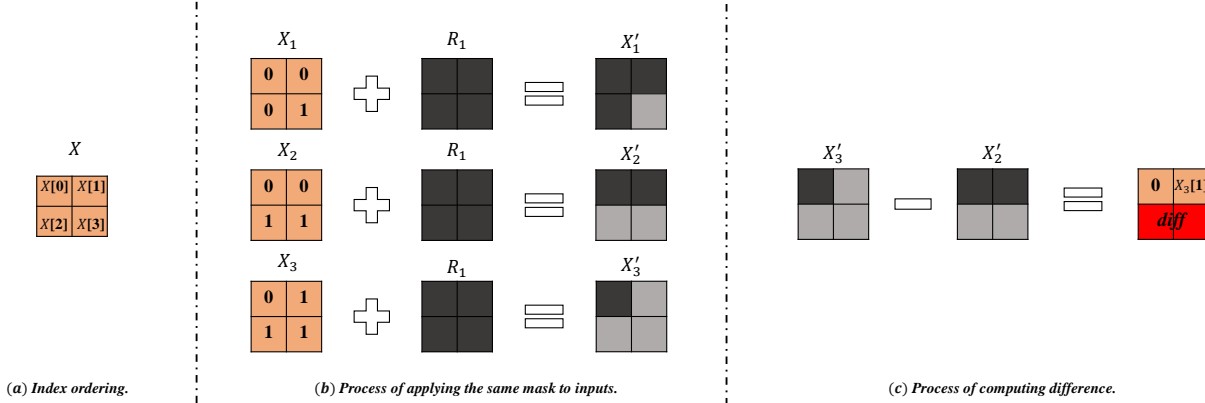

(a) Index ordering.  (b) Process of applying the same mask to inputs.  (c) Process of computing difference.

Figure 2: The four possible cases encountered when analyzing a collision triplet. Orange blocks denote the true values (0 for zero, 1 for positive), black blocks correspond to the mask, gray blocks represent masked positive values, and red blocks indicate the differences of the true values.

**Four Cases.** Figure 2 illustrates the four input blocks, each of which represents one of the four possible cases identified in the analysis. Without loss of generality, when discussing each case, we assume that for each component $i$, the values satisfy $X_3[i] \geq X_2[i] \geq X_1[i]$. The input can fall into one of the following four cases:

1. **Case 1:** All three true values are 0, corresponding to the first input block in Figure 2(b). In this case, only the values of the mask are observable, which provides no useful information for parameter recovery.

2. **Case 2:** Two components, $X_1[1]$ and $X_2[1]$, are zero, while the corresponding component $X_3[1]$ is positive. In this case, after applying the mask, the values remain two identical and one different. The two identical values correspond to the mask itself, as illustrated by $X'_1[1]$, $X'_2[1]$ in Figure 2(b), while the value of $X'_3[1]$ is equal to the sum of the true value $X_3[1]$ and the mask. At this point, we can eliminate the mask by computing the difference, thereby recovering the true value $X_3[1]$.

3. **Case 3:** One component $X_1[2]$ is zero, while the other two components $X_2[2]$ and $X_3[2]$ are positive. In this case, $X'_1[2]$ corresponds to the mask value, whereas both $X'_2[2]$ and $X'_3[2]$ equal their true values plus the mask value. The difference between $X'_2[2]$ and $X'_3[2]$ directly yields the difference between the corresponding true values. Furthermore, the differences $X'_2[2] - X'_1[2]$ and $X'_3[2] - X'_1[2]$ recover the true values $X_2[2]$ and $X_3[2]$, respectively.

4. **Case 4:** All components $X_1[3]$, $X_2[3]$, and $X_3[3]$ are positive. In this case, the masked values $X'_1[3]$, $X'_2[3]$, and $X'_3[3]$ each equal their corresponding true values plus the mask value. Their pairwise differences directly yield the corresponding true differences.

In practice, **Case 1** cannot be utilized for parameter recovery, since the observed values contain only the mask itself. By contrast, **Case 2** can be directly exploited to recover the parameters, as the differences effectively eliminate the mask and reveal the true values. Moreover, the availability of non-zero values enables the recovery of not only the coefficient matrix but also the bias terms.

For **Case 3** and **Case 4**, the attacker cannot distinguish between the two, since after applying the mask all three values become different from one another. At this stage, the sole certifiable observation is that the two largest masked values correspond to activations that were strictly positive before the mask was applied. Their differences correspond directly to the differences of the underlying true values.

## 3.2 First-Round Parameter Recovery

As the attacker is able to control the input of the neural network, the input mask of the first layer can be ignored. At this stage, when collecting collision triplets, we only need to focus on the outputs of the first-round ReLU, which serve as the inputs to the second-round linear layer. We use the process of recovering the first-round parameters as an example to illustrate our attack methodology. The procedure of attacking the first round of the neural network is as follows.

1. **Random input generation.** Randomly generate $n$ inputs and queries the neural network.
2. **Collision collection.** Collect collision triplets from the first ReLU outputs (i.e., the inputs to the second convolutional layer), denoted as $(y_1, y_2, y_3)$, and record their corresponding inputs $(x_1, x_2, x_3)$.
3. **Equation construction.** For each component of $y_1, y_2, y_3$, denoted as $y_1[i], y_2[i], y_3[i]$, construct linear equtaions relating the unkonwn network parameters.
   - **Scenario 1 (all distinct):** If $y_1[i], y_2[i], y_3[i]$ are all different, select the largest two values. For example, if $y_1[i] > y_2[i] > y_3[i]$, compute the differences
$$\Delta y = y_1[i] - y_2[i], \ \ \Delta x = x_1[i] - x_2[i], \tag{1}$$
   and use $(\Delta x, \Delta y)$ as one linear equation relating the unknown convolution and batch normalization parameters.
   - **Scenario 2 (two identical, one distinct):** If two outputs are identical and the third differs, e.g.,$y_1[i] > y_2[i] = y_3[i]$, compute
$$\Delta y = y_1[i] - y_2[i], \tag{2}$$
   and use $(x_1[i], \Delta y)$ to form an equation involving both the weights and bias.
4. **Iteration until full rank.** Repeat Steps 1-3 to collect sufficient equations, until the coefficient matrix reaches full rank(i.e., its rank equals the total number of unknown parameters).

Here, Scenario 1 corresponds to the previous Case 2, while Scenario 2 corresponds to the previous Cases 3 and 4.

It is worth noting that both the convolution and batch normalization operations are linear. Therefore, what we recover here are their composite parameters.

For convolutional layers, since ReLU activations occur with a certain probability, we can collect data from the above two cases to construct equations for solving the composite weights and biases of the convolution and batch normalization operations. However, for the fully connected (FC) layers in the output stage, since no ReLU follows them, only the weights can be recovered when only differential information is available.

Under the ideal assumption, the probability of a ReLU activation being positive is 50%. During the inference process of a convolutional neural network, each convolutional kernel a is applied multiple times across different spatial locations. The number of parameters is usually much smaller than the number of activations. Therefore, for most layer, only one or two triplets are are enough to generate a sufficient set of linear equations, enabling the recovery of all channel-wise convolution kernel parameters in that layer. For layers with smaller output dimensions and a larger number of channels (which implies fewer available equations but more unknowns), more collision triplets are required.

## 3.3 Recovery of Subsequent Linear Layers

In section 3.2, We illustrate the process of recovering neural network parameters via differential attacks using the first linear layer as an example. The adversary has an advantage when recovering the first layer: it can control the inputs, meaning that the input masks are effectively nonexistent at this stage. At this point, the attacker only needs to gather collision triplets from the outputs, i.e., the inputs to the subsequent linear layer. For recovering later layers, the same inputs are passed through the already recovered first-layer parameters to obtain the corresponding outputs, i.e., the inputs to the second layer. Figure 3 marks the intermediate states necessary for recovering the second layer.

The attacker simply collects collision triplets from the second-round outputs and applies a differential attack to recover the parameters. The attack on these subsequent layers follows exactly the same procedure as that used for the first layer.

## 3.4 Detecting Collision Triplets

In TEE-based protection schemes, the outputs of ReLU functions are masked. Since ReLU activations are zero with a certain probability and the outputs are high-dimensional vectors, two inputs protected by the same mask will exhibit

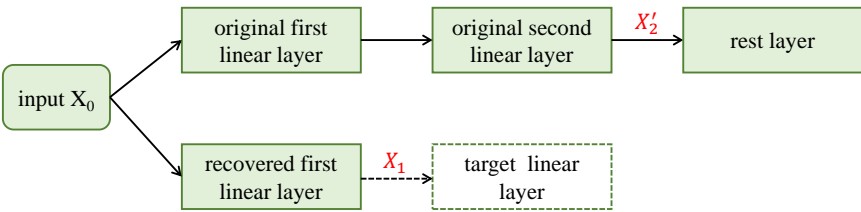

Figure 3: Intermediate states for recovering the 2nd linear layer parameters, where $X_1$ is the known input the second layer and $X_2'$ denotes masked output of the second layer.

many identical components in their masked outputs. This property enables the detection of mask reuse across different inputs. If two outputs share identical values in certain components, this indicates that they are protected by the same mask.

**Rationality Analysis.** When the outputs of two ReLU functions are masked with different masks, all of their components are completely random. We assume that the data type used in the neural network is a 32-bit floating point. By treating the intermediate states as random variables, their value space can be considered as $2^{32}$, given the 32-bit floating-point representation. The probability that any component of the two masked outputs coincides is $2^{-32}$.

Under mask reuse, the probability of coincidence for a particular component is $p_z^2$, since both ReLU outputs must simultaneously evaluate to zero prior to masking. Here, $p_z$ denotes the probability that a ReLU activation equals zero. $p_z^2$ is typically much greater than $2^{-32}$.

**Data volume for collecting collision triplets.** In order to collect collision triplets for subsequent analysis, it is necessary to acquire a sufficient number of inputs to ensure that triple collisions occur with high probability. This problem can be reduced to the generalized birthday problem(Wagner, 2002). Considering a mask space of size $n$, the expected number of samples required to observe a triple collision scales as $2^{\frac{2n}{3}}$, according to the generalized birthday paradox.

## 4   PATCH TO THE MASKING SCHEME: COMBINED MASK

Existing TEE-based model protection schemes must prepare a set of masks in advance for each linear layer. Once these masks are exhausted, security issues arises. The preparation of masks must also be performed inside the TEE. Specifically, for each linear layer, an input mask has to be generated, and the corresponding output mask must be computed. This process is computationally intensive.

As presented in Section 3, our method for detecting collisions involves finding two distinct inputs that produce identical values at multiple spatial locations within the same layer. Such occurrences are interpreted as the inputs having been processed with the same mask. Therefore, to achieve a secure masking, it is necessary to reduce the correlation. In this section, we introduce a new masking scheme that breaks the internal dependencies of the mask, thereby enhancing security while also reducing the computational complexity of the preprocessing stage.

For a linear layer, a distinct mask value is generated for each component of the input mask, and the output corresponding to each mask value is computed. These outputs represent partial contributions to the overall output mask, rather than the complete mask itself. The specific computation procedure is described as follows.

$$\begin{bmatrix} w_{11} & w_{12} & w_{13} \\ w_{21} & w_{22} & w_{23} \\ w_{31} & w_{32} & w_{33} \end{bmatrix} \cdot \begin{bmatrix} r_1 \\ r_2 \\ r_3 \end{bmatrix} = r_1 \begin{bmatrix} w_{11} \\ w_{21} \\ w_{31} \end{bmatrix} + r_2 \begin{bmatrix} w_{12} \\ w_{22} \\ w_{32} \end{bmatrix} + r_3 \begin{bmatrix} w_{13} \\ w_{23} \\ w_{33} \end{bmatrix}$$

Here, $r_1$, $r_2$, and $r_3$ denote the input masks that are independently generated at random. $w_{ij}$ are the parameters of the linear layer. We can generate multiple mask fragments for each component of the input and precompute their corresponding outputs. For each inference, a mask fragment is randomly selected from these precomputed outputs to form a new random mask.

Compared to the previous masking scheme, this approach can generate more masks with the same computational cost. This significantly enhances the security of the model.

## 5 EXPERIMENTS

In this section, we present the results of the differential on custom neural networks. In our experiments, we use a custom residual network, MiniResNet. It comprises three residual stages with channel configurations of [16, 32, 64], followed by an adaptive average pooling layer and a fully connected layer with 100 output units. For simplicity, we adjust the activation probability of each layer's ReLU to be very close to 50%. The architecture of the model is shown in Figure 4.

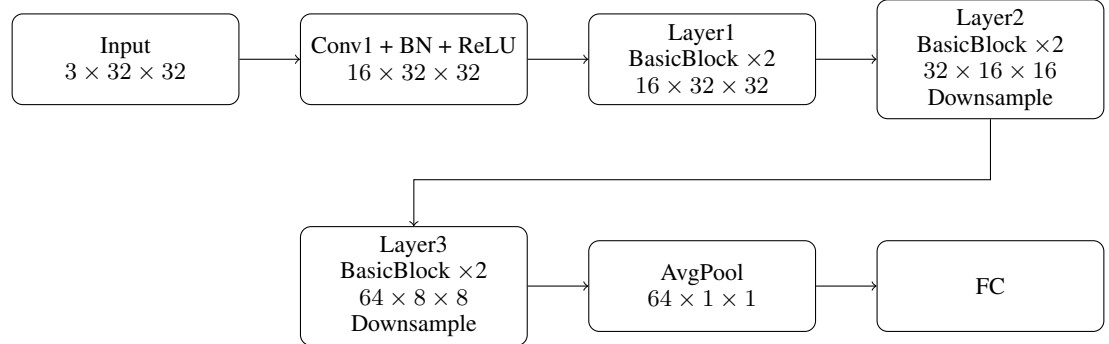

Figure 4: MiniResNet network architecture.

**Remarks on Data Collection and Linear System Solving.**  For a given linear layer, although a small amount of data (e.g., one collision triplet) is sufficient to construct enough linear algebraic equations to solve for the corresponding layer's parameters. However, the resulting linear system may be ill-conditioned, making the solution highly sensitive to noise or numerical errors in the input and output data. To mitigate this issue, we adopt two complementary strategies. First, during data collection, we incrementally construct the equation matrix by selectively adding input-output samples that increase the matrix rank, thereby ensuring linear independence. We further monitor the condition number of the updated matrix and discard samples that cause excessive growth, ensuring that the resulting system remains both full-rank and well-conditioned. Second, when solving the linear system, we employ $\ell_2$-regularized least squares (Ridge regression):

$$\hat{W} = \arg\min_{W} \|AW - b\|_2^2 + \lambda\|W\|_2^2,$$

where $A$ is the constructed equation matrix, $b$ is the response vector, and $\lambda$ is a small regularization parameter. The first term on the right-hand side fits the data, while the second term penalizes overly large weights, preventing the parameters from diverging. The regularization term improves numerical stability and ensures robust recovery of the layer parameters even if the system is nearly ill-conditioned.

Nevertheless, even with these strategies, the recovered parameters may still differ from those of the original network. Achieving more accurate recovery requires collecting additional input-output data, as confirmed by our experiments.

In this experiment, the output of the FC layer serves directly as the network output (and is therefore observable to an attacker), allowing the attacker to recover the parameters of all linear layers in the network. We further evaluate this process by recovering the parameters under different amounts of data and comparing the quality of the recovered parameters. We performed four sets of experiments to examine how data constraints affect parameter recovery: the first set imposed no restriction on the number of available samples (using the minimum amount of data), the second required that at least 10 collision triplets be used for each channel, the third increased this requirement to at least 20 collision triplets per channel, and the fourth further increased it to at least 30 collision triplets per channel. The recovery results of these three experiments are summarized in Table 1.

| Experiment Number | Data Constraint | Similarity Rate |
|---|---|---|
| Exp. 1 | No restriction | 32.33% |
| Exp. 2 | $\geq$ 10 collision triplets / channel | 79.16% |
| Exp. 3 | $\geq$ 20 collision triplets / channel | 98.72% |
| Exp. 4 | $\geq$ 30 collision triplets / channel | 99.63% |

Table 1: Parameter recovery results under different data constraints.

For the first eight linear layers, the system can be solved without additional constraints, as at most two collision triplets per channel are sufficient to construct a full-rank matrix. However, for the subsequent layers, the situation becomes more challenging due to the reduced output size and the increased number of channels, which jointly imply fewer equations can be collected while the number of unknowns grows. As a result, some channels require substantially more collision triplets for recovery; in particular, for the last convolutional layer, certain channels demand more than 100 collision triplets to achieve a solvable system.

Additionally, for each experimental setting, we computed the mean squared error (MSE) of the intermediate activations before the ReLU nonlinearity (pre-activation) between the original network and the recovered network using the same inputs. The corresponding experimental results and statistics are shown in Figure 5.

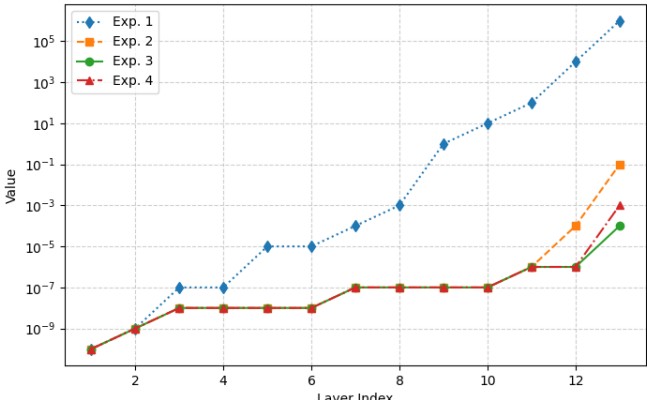

Figure 5: MSE of intermediate states under different data constraints.

From Table 1 and Figure 5, it can be observed that, with a sufficient amount of data, our attack is able to recover the parameters of the target network's linear layers with high accuracy, particularly for the shallower layers that have larger output feature dimensions. For layers with smaller output feature dimensions, a larger amount of data is required to accurately recover their parameters.

## 6 DISCUSSION

In our attack experiments, we primarily targeted the FC layer, which directly exposes the network's output and thus leaks more information to the attacker. In practical scenarios, however, the FC layer is typically followed by a softmax layer, and only the final predicted class is observable. This additional post-processing inevitably leads to information loss, making the recovery of FC layer parameters considerably more challenging.

To mitigate this difficulty, the attack can be extended into a two-stage process that combines our analytical method with data-driven training. In the first stage, a subset of model parameters is analytically recovered. In the second stage, the remaining parameters are refined through training (e.g., fine-tuning or optimization) using collected data, effectively completing model reconstruction. Our preliminary attempts along this direction indicate that partially recovered parameters indeed provide a better starting point compared to random initialization. Nevertheless, without advanced training strategies, the reconstructed models tend to converge to poor local optima, resulting in limited similarity.

Addressing this limitation remains an important direction for future research. We believe that integrating more sophisticated optimization techniques and training heuristics will further enhance the effectiveness of our method and bridge the gap between theoretical feasibility and practical applicability.

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

# A USE OF LLMs

In this work, the Large Language Model (LLM) was employed exclusively for grammar and spelling checking of the manuscript. The LLM was not involved in generating new content, formulating ideas, or conducting data analysis. Its role was limited to assisting in language polishing in order to improve readability and ensure clarity of expression.

