# OpenReview forum: "Attacking and Securing Masking Scheme for TEE-Based Model Protection"
_ICLR.cc/2026/Conference — ICLR 2026 Conference Withdrawn Submission_

### Official Review · Reviewer_pCGc · 2025-10-28

**Soundness:** 1
**Presentation:** 1
**Contribution:** 1
**Rating:** 0
**Confidence:** 5

**Summary:**

This paper significantly alters the margins, violating the official ICLR 2026 formatting guidelines.

**Strengths:**

N/A

**Weaknesses:**

This paper significantly alters the margins, violating the official ICLR 2026 formatting guidelines.

**Questions:**

N/A

---

### Official Review · Reviewer_16yQ · 2025-10-31

**Soundness:** 3
**Presentation:** 4
**Contribution:** 2
**Rating:** 4
**Confidence:** 4

**Summary:**

The paper studies the weakness of existing TEE-based model masking schemes, presents a differential attack that can recover model parameters with high accuracy, and proposes a new combined masking scheme that generates more secure mask variants with the same cost.

**Strengths:**

1) Attack design is concrete and reproducible, showing strong analytical reasoning using collision triplets.
2) Well-written manuscript
3) Empirical validation (98% recovery) is convincing and well presented.

**Weaknesses:**

1) The threat model is too simplified as it assumes a fully trusted TEE and ignores side-channel or partial leakage.
2) The attack success is limited to shallow networks and deeper architectures only work after retraining.
3) Comparisons to prior TEE defenses (e.g., NNSplitter) are minimal
4) No runtime overhead analysis.

**Questions:**

1) The paper provides a valuable contribution by exposing a clear vulnerability in precomputed masking used by TEE-protected inference systems.
2) However, the scope of evaluation is narrow, by focusing only on one small CNN. There’s no test on real frameworks like SGX/TrustZone nor a performance comparison for the new masking scheme.
3) I also suggest adding overhead analysis of the proposed method too, and demonstrate its effectiveness in presence of recent TEE defenses.
4) The paper is well written and logically structured. Minor typos (“adersary”, “unkonwn”) should be fixed.

---

### Official Review · Reviewer_TcxH · 2025-11-09

**Soundness:** 1
**Presentation:** 1
**Contribution:** 1
**Rating:** 0
**Confidence:** 5

**Summary:**

This paper seriously violates the ICLR formatting rules by using reduced margins, which allows the authors to include more content than permitted. This creates an unfair advantage compared to other submissions. I recommend a desk rejection.

**Strengths:**

NA

**Weaknesses:**

NA

**Questions:**

NA

---

### Note · Authors · 2025-11-17

I have read and agree with the venue's withdrawal policy on behalf of myself and my co-authors.